# Probiotic Encapsulation: Bead Design Improves Bacterial Performance during In Vitro Digestion

**DOI:** 10.3390/polym15214296

**Published:** 2023-11-01

**Authors:** Yesica Vanesa Rojas-Muñoz, Patricio Román Santagapita, María Ximena Quintanilla-Carvajal

**Affiliations:** 1Maestría en Diseño y Gestión de Procesos, Facultad de Ingeniería, Campus Universitario del Puente del Común, Universidad de La Sabana, Chía 250001, Colombia; yesicaromu@unisabana.edu.co; 2Departamento de Química Orgánica, Facultad de Ciencias Exactas y Naturales, Universidad de Buenos Aires & Centro de Investigación en Hidratos de Carbono (CIHIDECAR, UBA-CONICET), Buenos Aires 1428, Argentina; patricio.santagapita@qo.fcen.uba.ar; 3Grupo de Investigación de Procesos Agroindustriales (GIPA), Facultad de Ingeniería, Campus Universitario del Puente del Común, Universidad de La Sabana, Chía 250001, Colombia

**Keywords:** probiotic, encapsulation, ionotropic gelation, functional food, INFOGEST

## Abstract

The stability and release properties of all bioactive capsules are strongly related to the composition of the wall material. This study aimed to evaluate the effect of the wall materials during the encapsulation process by ionotropic gelation on the viability of *Lactobacillus fermentum* K73, a lactic acid bacterium that has hypocholesterolemia probiotic potential. A response surface methodology experimental design was performed to improve bacterial survival during the synthesis process and under simulated gastrointestinal conditions by tuning the wall material composition (gelatin 25% *w*/*v*, sweet whey 8% *v*/*v*, and sodium alginate 1.5% *w*/*v*). An optimal mixture formulation determined that the optimal mixture must contain a volume ratio of 0.39/0.61 *v*/*v* sweet whey and sodium alginate, respectively, without gelatin, with a final bacterial concentration of 9.20 log_10_ CFU/mL. The mean particle diameter was 1.6 ± 0.2 mm, and the experimental encapsulation yield was 95 ± 3%. The INFOGEST model was used to evaluate the survival of probiotic beads in gastrointestinal tract conditions. Upon exposure to in the vitro conditions of oral, gastric, and intestinal phases, the encapsulated cells of *L. fermentum* decreased only by 0.32, 0.48, and 1.53 log_10_ CFU/mL, respectively, by employing the optimized formulation, thereby improving the survival of probiotic bacteria during both the encapsulation process and under gastrointestinal conditions compared to free cells. Beads were characterized using SEM and ATR-FTIR techniques.

## 1. Introduction

Currently, consumer awareness is increasingly interested in the health impact of food consumption [1]. Consequently, some foods have added bioactives in their matrix, potentially reducing the risk of suffering specific diseases and providing better health benefits. This is the definition of functional foods. Some examples include products enriched with vitamins, minerals, or fiber, or supplemented with probiotics [2,3]. The most widely accepted definition of a probiotic is “live microorganisms that, when administered in adequate amounts, confer a benefit for the health of the host” [4,5]. The global probiotics market size was estimated at USD 40 billion in 2017, and it is projected to reach USD 65.87 billion by 2024 [6].

The most used probiotics are lactic acid bacteria (LAB) of the *Lactobacillus* and *Bifidobacterium* genera [7,8]. *Lactobacillus fermentum* K73 is a LAB isolated from fermented food consumed on the Colombian Atlantic coast. According to studies performed in vitro, *L. fermentum* has probiotic and hypocholesterolemic potential [9]. Probiotic foods are required to have a minimum of 1 × 10^6^ CFU/g of viable bacteria until the end of their shelf life [10]. However, producing probiotic foods on a large scale presents a significant challenge as probiotic strains require special handling and production methods to maintain their viability and functionality. The production process can potentially damage the probiotic cells, affecting the final concentration of viable bacteria in the food product [11]. In addition, for probiotics to be effective, they must survive the harsh physiological conditions of the gastrointestinal tract [12,13]. Therefore, ensuring the viability and functionality of probiotic cells in functional food is critical for the food industry. To overcome this challenge, one strategy is the encapsulation of cells by using a carrier system that protects and releases them at the site of action [14,15].

Currently, the main encapsulation technologies for probiotics in the food industry are extrusion, emulsion, and spray-drying. The extrusion technique has been well received by the industry due to the benefits it brings at low cost, such as the avoidance of elevated temperatures or organic solvents, and most of the materials that can be used being GRAS (generally recognized as safe) [16]. In its simplest form, the encapsulation process is based on a gelling solution of a biopolymer mixed directly with the bioactive system of interest, which is then added into a gelling solution to form beads via a crosslinking reaction between the ion and the polyelectrolyte [17,18]. Wall materials are often polymers that aim to separate the internal phase from the surrounding matrix. There is a wide spectrum for the selection of food-grade polymers suitable for application in microencapsulation by extrusion, among which are sodium alginate, chitosan, milk proteins, and gums [15,19]. Sodium alginate is a linear anionic and hydrophilic biopolymer of natural origin. Its compatibility, biodegradability, and low-cost characteristics have made it the most widely used material in the extrusion technique. It is composed of blocks of β-D-mannuronic acid (M) and α-L glucuronic acid (G). In solution, sodium alginate forms a cross-linked structure in which its anionic acid groups can react with divalent (Zn^2+^, Ca^2+^, Ba^2+^, and others) or polyvalent cations, forming alginate hydrogels with a structure commonly known as “egg-box” [20,21]. 

The development of Ca-alginate beads for encapsulating bioactive has been a major research focus. However, these beads have several drawbacks, including high porosity, susceptibility to acidic environments, and scaling problems. To address these issues and optimize the encapsulation process, researchers have developed beads by incorporating other polymers to achieve a synergistic effect [22]. In previous studies in which probiotics have been encapsulated using this technique, the unitary operation of centrifugation has been used to concentrate bacterial biomass, which is later mixed with encapsulating materials [22,23,24]. However, this operation increases production costs, making it challenging for the food industry to scale up production of functional foods based on probiotics. This study presents a novel approach where the whey culture medium in which *L. fermentum* K73 grows is used as the bead wall material, eliminating the need for centrifugation [25].

The objective of this work was to evaluate the effect of the formulation of wall materials in the extrusion encapsulation process on the viability of *Lactobacillus fermentum* K73, improving the performance of the bacterial counts in their passage through the simulated model of gastrointestinal conditions (INFOGEST). The results obtained from the optimal mixture response surface methodology (RSM) allowed for the selection of the volume ratio of each wall material evaluated (type A gelatin, sweet whey, sodium alginate) in the optimal formulation, and the beads were subsequently characterized. This approach represents a significant advance in the development of encapsulation techniques for probiotics, with potential applications in functional foods. 

## 2. Materials and Methods

### 2.1. Materials

The materials used in the present study were de Man Rogosa and Sharpe (MRS) agar, MRS broth, peptone water, and yeast extract obtained from Scharlau Microbiology (Barcelona, Spain). Glycerol and di-hydrated calcium chloride (CaCl_2_·2H_2_O) were purchased from PanReac AppliChem (Barcelona, Spain). Sodium alginate (M/G 0.9; MW 1.40 × 10^4^ g/mol) and the enzymes used in the in vitro digestion assays (pancreatin P7545, lipase L3126, and ox-bile extract 70168) were acquired from Sigma-Aldrich (Saint Louis, MI, USA). Sweet whey (crude protein 11%, crude fat 1.5%, and lactose 61%) was obtained from Saputo Ingredients (Lincolnshire, IL, USA), and gelatin (type A, Bloom 270 g, and MW~100 kDa) was obtained from Cimpa S.A.S. (Bogotá, Colombia).

### 2.2. Methods

#### 2.2.1. Strain and Bacterial Conservation

The strain *L. fermentum* K73 used in this work was obtained from the collection Usab-Bio of the Department of Engineering, Universidad de La Sabana (Colombia). To preserve the stock cultures, 2 mL vials containing MRS broth and glycerol as a cryoprotective agent, at 40% (*v*/*v*) in a volume ratio of 1:1, were stored at −80 °C (Ultra-freezer, Precisa, Hangzhou, China). Prior to use, the bacterial culture was incubated in MRS broth for 12 h at 37 °C under static aerobic conditions [9].

#### 2.2.2. Biomass Production

Biomass production was carried out in a 1.3 L bioreactor (BioFlo 110; New Brunswick Scientific Co., Enfield, CT, USA). Operational conditions were the following: temperature at 37 °C and agitation speed of 100 rpm for 10 h. The culture medium was prepared with 8% (*w*/*v*) sweet whey and 0.22% (*w*/*v*) yeast extract adjusted to a final pH of 5.5 with 1 M HCl. After sterilizing the culture medium at 121 °C for 15 min, *L. fermentum* K73 was inoculated at 10% (*v*/*v*) [25].

### 2.3. Formulation of Wall Materials

#### 2.3.1. Preparing Solutions

Sodium alginate 1.5% (*w*/*w*) and gelatin type A at 25% (*w*/*w*) solutions were prepared with deionized water at 90 °C for 30 min under magnetic stirring (250 rpm) [24,26]. Calcium chloride hardener solution (100 mM) was prepared with deionized water, and it was adjusted to a final pH of 4.5 according to preliminary assays. All solutions were sterilized.

#### 2.3.2. Mixture Design

The experimental design for the formulation of the bead wall materials was performed using an Optimal Mixture response surface methodology (RSM) through Design-Expert software version 11.0.0 (Stat-Ease Inc., Minneapolis, MN, USA). The design included 16 runs (as shown in Table 1) with 5 replicas. The response variable was viability (log_10_ CFU/mL). The following numerical factors were wall materials volume ratios: sodium alginate (0.40–0.90), gelatin type A (0.00–0.50), and sweet whey (0.00–0.50). Each mixture was inoculated at 20% (*v*/*v*). The ranges of the proportion of each material were selected according to preliminary evaluations.

The RSM was used to determine the optimal mix of bead wall materials while considering the maximization of probiotic viability (results are included in Table 1). The suggested optimal mixture was replicated in the laboratory, and the error between the predicted and observed response variable was calculated using Equation (1).
(1)$$\%Error=\frac{Predicted\;variable-Observed\;variable}{Predicted\;variable}\times 100$$

### 2.4. Probiotic Encapsulation

The encapsulation process consisted of loading the mixture into a syringe with a 0.6 mm diameter needle. The resulting string of drops was dropped into the hardener solution at a height of 3 cm. The instantly formed beads were left stirring in a CaCl_2_ solution with gentle magnetic agitation for 30 min to promote efficient gelling of the particles and the resulting beads were rinsed with distilled water [14].

#### 2.4.1. Cell Release

The release of the probiotic cells from the bead matrix was achieved using the technique documented by (Bevilacqua et al., 2020 [27]). Briefly, 5.0 g of beads was mixed with a 5.0% (*w*/*v*) sodium citrate solution at a dilution ratio of 1:10 (*v*/*v*) and homogenized using a vortex VG 3 (IKA, Werke, Germany) at 800 rpm for 60 s until the beads were completely dissolved. Subsequently, cell viability was determined using the plate count method.

#### 2.4.2. Cell Viability

The cell count of *L. fermentum* K73 for all the proposed experiments was carried out by making 1:10 (*v*/*v*) serial dilutions in peptone water 0.1% (*w*/*v*). Plating was performed on MRS agar from dilution 1 to 7 in triplicate. Incubation conditions were 37 °C for 24 h under aerobic conditions. Plates containing up to 250 colonies were numbered. The result was reported as the logarithm of the final colony-forming units per milliliter (CFU/mL).

#### 2.4.3. Encapsulation Efficiency (EE)

Cell entrapment efficiency was calculated as the difference between viable cell count before and after the microencapsulation process (Graff et al., 2008 [28]).
(2)$$EE\%=(log{(CFU/mL)}_{N_{2}}/{log(CFU/mL)}_{N_{1}})\times 100$$
where *N*_1_ and *N*_2_ indicate the number of viable bacteria in the mixture and released from the beads, respectively. 

### 2.5. Characterization of Beads

#### 2.5.1. Attenuated Total Reflectance–Fourier Transform-Infrared Spectroscopy (ATR-FTIR)

Fourier Transform Infrared (FTIR) spectra were acquired using a spectrometer (Varian model 630-IR, Agilent-Tech Inc., Santa Clara, CA, USA), with 16 scans collected from 675 to 4000 cm^−1^ at 4 cm^−1^. All samples were studied using a single reflection ATR system (Cary 630 ATR-TIR Instrument, Agilent-Tech Inc., Santa Clara, CA, USA) of Ge crystal and with an incident angle of 45°. Control reagents (sodium alginate and prepared culture medium) and beads were freeze-dried prior to measurement. 

Data were processed by using the free-license Spectragryph v1.2.15 software (developed by Dr. Friedrich Menges, Oberstdorf, Germany). Spectra were baseline-corrected (adaptative correction; coarseness: 50; offset 10), and were normalized between 0 and 1 and smoothed (Savitzky-Golay, 15 points, 2) for figure presentation. The FWHM (full width at half maximum) of the peak was calculated by using the software tool.

#### 2.5.2. Scanning Electron Microscopy (SEM)

Some morphological parameters of the beads were observed by scanning electron microscopy (FE-MEB LYRA3 Tescan, Bollebergen, Belgium). Samples were subjected to freeze-drying and air-drying to compare bead morphological changes between drying methods. For the freeze-drying process, beads were washed and frozen at −80 °C for 24 h, then were placed in the freeze-dryer (Labconco, FreeZone 4.5L, Fullerton, CA, USA) at −40 °C and 0.05 mbar pressure for 24 h. For the air-drying process, beads were ventilated in an air chamber at 40 °C for 4 h. Dried beads were stored in desiccators over phosphorus pentoxide (P_2_O_5_) for two days and were coated with gold before observation. SEM was operated at 10 kV and 60–5000× magnification. In addition, wet beads were observed using a high-performance scanning electron microscope (JSM-6490LV, JEOL, Zaventem, Belgium) in low vacuum mode.

#### 2.5.3. Particle Size

One hundred fresh beads were randomly selected and placed on dark paper, and a digital camera (Canon EOS 70D) was used to photograph them. The diameter of the beads was determined by processing the images in ImageJ software (V 1.50i, National Institutes of Health, Bethesda, MD, USA) [21].

### 2.6. Probiotic Cell Viability under INFOGEST Simulated Gastrointestinal Model

The INFOGEST protocol was used to simulate the conditions of the gastrointestinal tract proposed by [29,30]. Briefly, the oral phase was prepared by mixing 5 g of the sample (beads) and 5 mL of simulated salivary fluid (SSF). The final oral phase pH was brought to 7.0 and was shaken (100 rpm) at 37 °C for 2 min. Then, the gastric phase consisted of adding 10 mL of simulated gastric fluid (SGF) to the oral phase and adjusting it to a final pH of 3.0 with the required volume of HCl (1 M). The sample was under agitation (100 rpm) for 2 h at 37 °C. Finally, the intestinal phase was performed by mixing the gastric phase, 20 mL simulated intestinal fluid (SIF), and enzymes bile extract (10 mM), pancreatin (100 U/mL), and pancreatic lipase (2000 U/mL). The final pH of the intestinal phase was adjusted to 7.0, and it was incubated under the same conditions as the gastric phase. After each phase, a viability assay was carried out according to the 2.4.1 numeral. Bacteria survival was determined according to Equation (3).
(3)$$Survival\%=({log\left(CFU/mL\right)}_{final}/{log\left(CFU/mL\right)}_{initial})\times 100$$

### 2.7. Statistical Analysis

The significant test of the designs was performed by analysis of variance (ANOVA) with a confidence level of 95%. The coefficient of determination (R^2^) was used to evaluate the fit of the measurements to the regression models. Three replications were performed in all assays and the data were presented as the mean ± standard deviation. The RSM Optimal Mixture design was performed using the Design-Expert software (version 8.1.0, Stat-Ease Inc., Minneapolis, MN, USA). For the optimization of the response variable, the desirability criterion of the specialized software was used.

## 3. Results and Discussion

### 3.1. Formulation of Wall Materials: Mixture Design

The selection of suitable wall materials for the probiotic cell encapsulation process is closely related to the viability, stability, and release of the bioactive, since these materials protect the compound of interest from external factors, such as processing, storage, and conditions of the GI tract [15,31,32]. In the present study, the mixture’s optimal ratio between gelatin, sodium alginate, and sweet whey as bead wall materials was evaluated. The results of the experimental design of the mixture (Table 1) were adjusted to a statistically significant quadratic model (*p* ≤ 0.05). The coefficient of determination (R^2^) was 0.80, and the lack of fit of the model was not significant (*p* ≥ 0.05), as shown in Table 2. The analysis of variance (ANOVA) of the regression for the response variable “viability” showed two statistically significant double interactions between the sodium alginate and the other two materials. 

The second order polynomial equation in terms of the coded factors is:(4)Y=7.96A+8.46B+8.91C−3.31AB−2.98AC+0.18BC 
where Y is the predicted response variable (viability), and A, B, and C are the coded values of the proportion of alginate, gelatin, and whey in the mixture, respectively. 

Figure 1 shows the graphic optimization of the effect of the different wall materials on bacterial viability. The region between sodium alginate and sweet whey (A–C) shows a strong interaction with maximum viability. In the numerical optimization, alginate is present in a high proportion at the optimal point, and the alginate–whey mixture presents the maximum value of the response variable. This may be because increasing the proportion of alginate would allow for more Ca^2+^ ion binding sites and, consequently, for a greater number of alginate strands to be held together in the microcapsule structure, effectively protecting bacterial cells [33]. Moreover, an increasing sodium alginate concentration leads to higher retention of bioactive assuring the microstructural stability of the beads [34]. On the other hand, milk proteins are a wall material capable of forming gels. They can also have interactions with other polymers to form complexes, in this case with alginate [35]. The model obtained from the experimental mixture design allowed for selection of the optimal ratio for the formulation of the wall materials based on the desirability criterion. The optimal point was a mixture composed of sweet milk whey (0.604) and sodium alginate (0.396) with a predicted viability of 9.261 log_10_ CFU/mL. The optimal point was replicated experimentally as a validation run to evaluate the predictive capacity of the model, obtaining a low error rate (0.63%) considering the experimental viability of 9.20 log_10_ CFU/mL. This observation is consistent with the study of [36], in which the wall materials were studied individually (whey protein and alginate) and in a mixture, with the finding that the optimal proportion that maximized the viability of the microorganism was 0.62/0.38, respectively. Dehkordi et al., (2020) [37] investigated the increase in the viability of *L. acidophilus* encapsulated in alginate–sweet whey capsules compared to alginate–whey protein isolate (WPI) capsules. In their study, the cell suspension was centrifuged at 3000× *g* for 15 min, washed with saline solution, and then resuspended in deionized water. However, in the present article, this step is not required due to the novel composition of the *L. fermentum* culture medium used. Moreover, WPI contains 93% protein content, which can increase the final cost of the capsule. In contrast, sweet milk whey is an agro-industrial waste and could potentially be a more economical option.

In addition, the encapsulation efficiency of the extruded beads with the optimal mixture in this study was 94.8%, a value consistent with the existing literature. Donthidi et al. (2010) [38] reported that encapsulation with alginate and whey protein enhanced the survival of probiotics with a 90.9% encapsulation efficiency. In another study [39], microencapsulated *L. bulgaricus* cells in alginate–milk microspheres using vibratory technology reported a high encapsulation yield (99%).

It is interesting to highlight that gelatin was excluded from the optimal point by the model since its inclusion reduced the encapsulation efficiency at any of the assayed concentrations, even though the encapsulation efficiency values were rather high (as an absolute value). Gelatin has been used in the food industry as a vehicle for probiotic encapsulation in alginate–gelatin beads [40,41] and was an excellent excipient for the encapsulation of *Lactobacillus fermentum* K73 by electrospinning. However, gelatin-containing beads were more challenging to produce since the viscosity of gelatin changes with temperature, and since the probiotic bacteria is mesophilic, it was encapsulated with gelatin at 37 °C. The addition of secondary components and their interaction is critical to tune the fine structure of Ca(II)-alginate beads [42,43] since the interactions among them could affect the network, its pore size, and the interactions established within the wall materials the cells. The interaction among the materials and the characteristics of the beads are included in the next section.

### 3.2. Characterization of Beads

#### 3.2.1. Attenuated Total Reflectance–Fourier Transform-Infrared Spectroscopy (ATR-FTIR)

Figure 2A,B show the ATR-FTIR spectra of the beads and the wall materials. Considering the components of the beads, the culture medium (CM) shows the typical signal corresponding to sweet whey, which is composed of lactose and oligosaccharides, proteins, and fat [44]. Moreover, the symmetric and asymmetric stretching of carboxylic acids from sodium alginate at 1593 and 1405 cm^−1^ are also observed, respectively (Figure 2A). Prior to analyzing the spectrum of the beads, is important to keep in mind the type of interactions that can be established among the wall constituents. Considering the pH of the optimal mix solution prior to gelation (4.82), electrostatic interactions between sodium alginate (SA) and sweet whey (SW) are expected. The isoelectric point of SW main protein (β-lactoalbumin) is 5.1–5.2, given an overall positive charge in the protein, as also reported [45]. On the other hand, sodium alginate is negatively charged at a pH higher than 3.65 [46]. Several peaks shifted in the beads with respect to the plain components; the symmetric and asymmetric stretching of carboxylic acids changed from 1593 to 1598 and 1405 to 1422 cm^−1^, respectively, as shown in Figure 2B. There are also changes at 3000–3600 (OH stretching) in both position and relative intensities, revealing rearrangements of the hydrogen bonds (which change their average length) to 3235 from 3216 and 3267 cm^−1^ for SA or CM, respectively (Figure 2A). Moreover, the FWHM (full width at half maximum) of the peak of SW-ALG beads is thinner than those of SA or CM (380 vs. ~420 cm^−1^ for individual components), which is linked to a higher degree of homogeneities of the intermolecular interactions, which reduce the dispersion of the vibrational levels and higher conformational selectivity [47]. By comparing to sodium alginate (the main component), even though the increase in wavenumber indicates a reduction in hydrogen-bond density and strength, this change has also been linked to a decreased molecular packing, hence the greater protection in the dried state [47].

Moreover, some changes were also observed in the main peaks of the sugars, particularly for C-O-C and C-O-H of the sugar ring, showing several displacements between 1200 and 800 cm^−1^: the maximum (COC stretching) was shifted from 1025 and 1017 (of SA or CM, respectively) to 1011 cm^−1^ in beads; and the C-OH stretching from 1083 and 1068 to 1077 cm^−1^ (Figure 2B). Furthermore, a new peak at 939 from the bands at 947 and 933 cm^−1^ of CM and SA, respectively, and a shift from 989 (of CM) to 993 cm^−1^ in beads were observed (Figure 2B), accounting for more interactions (mainly hydrogen bonding and van der Waals forces) between components in the formulated beads.

#### 3.2.2. Particle Size and Morphological Characterization

The morphological characterization of the beads was performed immediately after encapsulation. Fresh beads are shown in Figure 3A. The encapsulated probiotics were characterized by performing image analysis. SW-SA beads showed spherical morphologies with an average diameter (±standard deviation) of 1.6 ± 0.2 mm and high circularity (0.8 ± 0.1). The obtained beads exhibited adequate mechanical stability under handling. Different authors have reported that beads loaded with probiotics have diameter values between 1.5 and 1.9 mm. However, these sizes vary widely depending on the encapsulation materials and extrusion diameter [16]. *Bifidobacterium longum* encapsulated by [48] in alginate–dairy matrices depicted varied sizes of microcapsules (2.3–3.1 mm) due to the type of encapsulation material. Lopes et al. (2017) [49] extruded alginate–gelatin beads to encapsulate *L. rhamnosus*, obtaining regular- and spherical-shape beads with a size ranging between 1.53 and 1.90 mm.

The morphology of the wet and dried beads was observed by SEM (Figure 3). Wet microcapsules showed a spherical shape without agglomerations and with a continuous surface without hollow areas or deep-crack morphology (Figure 3B), confirming particle size analysis. However, frozen and air-dried microcapsules slightly lose this spherical structure (Figure 3C,D, respectively), showing that the drying method affects the morphology of the beads. These results are in line with another study [43], which found that the loss of water determines the morphology of the final structure in each drying, as revealed by the contractions and small eruptions on the surface as a result of the massive loss of water (from 0.98 to 0.04 g H_2_O/g dry weight). The encapsulated *L. fermentum* K73 cells on the surface of the bead was directly observed to be covered by the matrix (Figure 3E). 

### 3.3. Probiotic Cell Viability under INFOGEST-Simulated Gastrointestinal Model

Probiotics play a significant role in human health by providing a protective effect on the microbiota in the gastrointestinal tract. Thus, ensuring the minimum dose at which the microorganism is capable of colonizing and exerting its beneficial activity is essential [50]. For this reason, the viability of free and encapsulated *L. fermentum* K73 cells was evaluated under the standardized static in vitro digestion protocol developed by the INFOGEST international network, as shown in Figure 4, for each GI tract phase. The viability of free probiotic cells significantly decreased (*p* ≤ 0.05) compared to encapsulated cells in SW-SA beads.

Survival rates showed no significant decrease in the oral phase of the two treatments evaluated, the viability of the encapsulated cells was 97.6% (0.21 log CFU/mL), and for free cells, it was 92.1%, (0.69 log CFU/mL) compared with the initial bacteria viability. After 2 h of incubation in the gastric phase (pH 3), free cells showed a significant decrease (67.2% or 2.87 cycles) compared with SW-ALG encapsulated cells (92.1% or 0.69 cycles). Encapsulation with alginate and sweet whey was effective in protecting probiotic cells since the reduction in the viability of the encapsulated bacteria after digestion was significantly lower (76.5%—1.91 log CFU/mL) than that observed in free cells (40.1%, 5.11 log CFU/mL). These results agree with previously reported studies, where alginate encapsulation in combination with other materials helped increase the viability of probiotic microorganisms after subjecting them to adverse conditions [51,52,53,54]. Eckert et al. (2018) [55] found that the non-encapsulated *Lactobacillus* spp. cells were sensitive to the conditions of the simulated gastrointestinal tract, and some even died after the gastric phase. However, the reduction in viability of whey–pectin–alginate probiotic microparticles during digestion was 2–3 log cycles, showing that encapsulation effectively protected cells by keeping viability above the requirements for probiotic foods. Lee et al. (2019) [17] encapsulated *L. acidophilus* KBL409 with alginate and chitosan, and observed a dramatic decrease in the survival rate (46.5%—4.95 log reduction) of free cells, while more than 80% of chitosan–alginate encapsulated cells were viable after digestion. 

In addition, it is important to point out that the count of viable encapsulated cells in the intestinal phase was 6.7 CFU/mL, which is higher than the limit required to classify a product as a probiotic (>6 log CFU/mL). This result highlights the protective effect of the optimal mixture of the beads under the digestion conditions. Considering the wall components and the established interactions among them (Figure 2), it seems that a favorable barrier against rapid degradation and/or diffusion at the periphery of the microspheres between whey–alginate may occur, protecting the microorganisms from the GI conditions and producing probiotic particles of high encapsulation efficiency.

## 4. Conclusions

In conclusion, the encapsulation of *Lactobacillus fermentum* K73 by ionotropic gelation using sodium alginate and sweet whey as wall materials was successful. The optimization process leads to a combination of two widely and well-known materials with the advantage of using the same culture medium as one of the wall materials. Bead design provided an additional barrier that led to obtaining a high viability and encapsulation yield of the probiotic (9.261 log_10_ CFU/mL, with a 94.8% of encapsulation efficiency). The particles produced with the optimal mixture exhibited improved structure and enhancements to probiotic toleration to simulated gastrointestinal conditions, conforming to the requirements for probiotic foods.

## Figures and Tables

**Figure 1 polymers-15-04296-f001:**
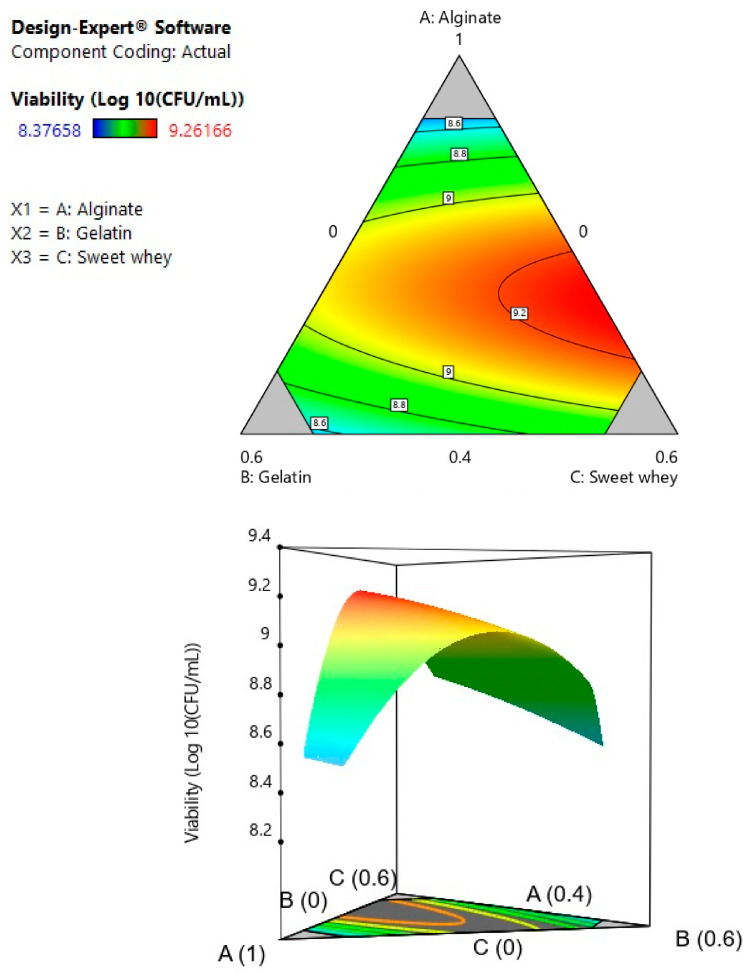
Contour plot and response surface for the variable viability of *Lactobacillus fermentum* K73.

**Figure 2 polymers-15-04296-f002:**
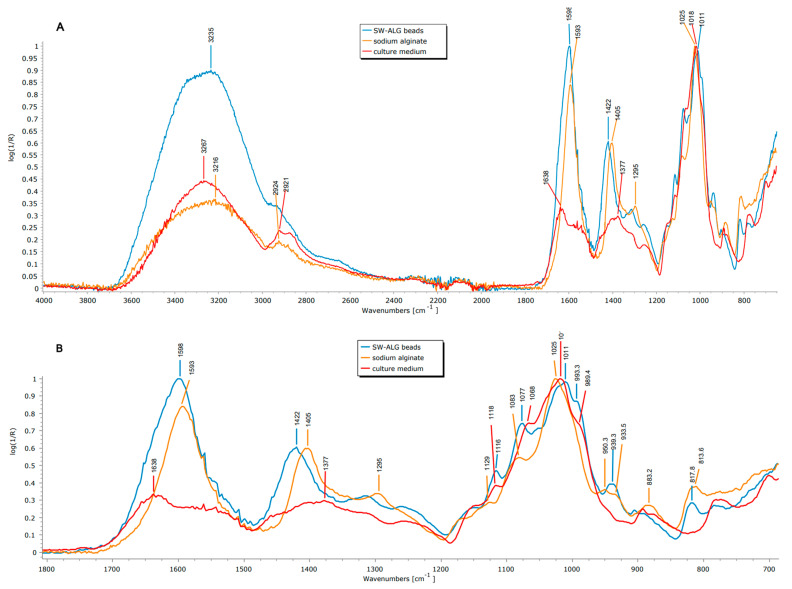
FTIR analysis of sodium alginate (ALG), culture medium—mainly sweet whey (SW)—and SW-ALG freeze-dried microcapsules: (**A**) full-range spectrum; (**B**) detailed region between 1700 and 700 cm^−1^.

**Figure 3 polymers-15-04296-f003:**
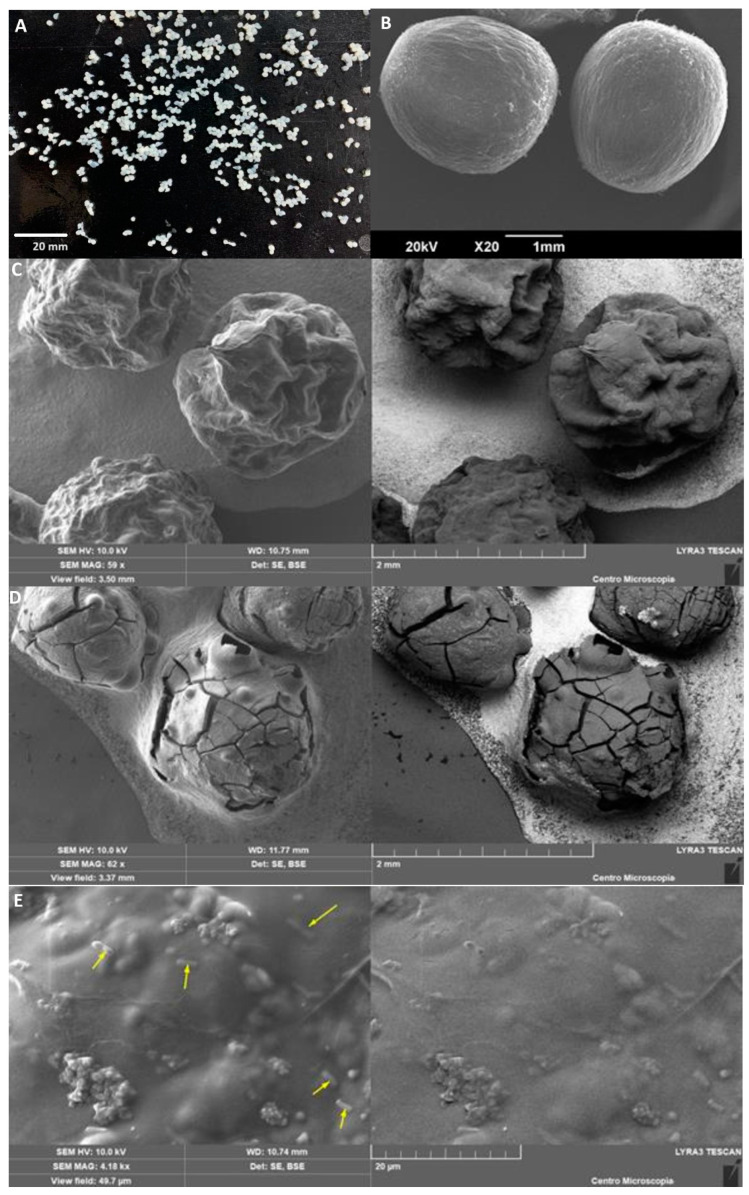
*L. fermentum* K73 encapsulated in (**A**) fresh beads, (**B**) wet beads, (**C**) freeze-dried beads, and (**D**) air-dried beads. (**E**) Bacteria on the surface of freeze-dried bead.

**Figure 4 polymers-15-04296-f004:**
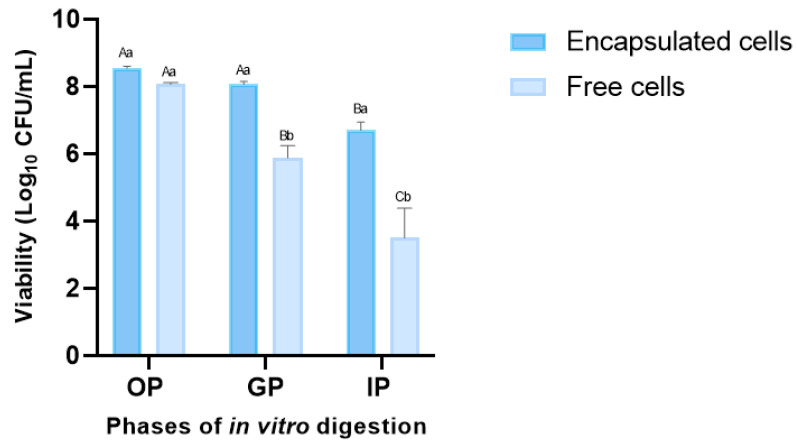
Viability of free and encapsulated *L. fermentum* K73 cells during OP (oral phase: 2 min, pH 7.0), GP (gastric phase: 120 min, pH 3.0), and IP (intestinal phase: 120 min, pH 7.0) according to the INFOGEST in vitro model. The mean value ± standard deviation of at least three independent measurements is included. (A–C) Different letters within the same treatment (encapsulated or free cells) indicate statistical significance between in vitro digestion phases (*p* < 0.05). (a,b) Different letters with the same in vitro digestion phase indicate statistical significance between treatments (encapsulated or free cells) (*p* < 0.05).

**Table 1 polymers-15-04296-t001:** Optimal bead wall material selection using a mixture experimental design.

Run	Factors			Response Variable
	A: Alginate	B: Gelatin	C: Sweet Whey	Viability (log_10_ CFU/mL)
1	0.80	0.02	0.18	9.12
2	0.50	0.26	0.24	9.05
3	0.55	0.00	0.45	9.11
4	0.40	0.10	0.50	8.91
5	0.66	0.34	0.00	9.14
6	0.90	0.06	0.04	8.47
7	0.50	0.26	0.24	8.95
8	0.52	0.37	0.11	9.18
9	0.66	0.34	0.00	9.04
10	0.40	0.50	0.10	8.38
11	0.65	0.15	0.20	9.05
12	0.40	0.10	0.50	8.91
13	0.76	0.18	0.06	8.87
14	0.50	0.26	0.24	9.04
15	0.68	0.00	0.32	9.26
16	0.40	0.50	0.10	8.65

**Table 2 polymers-15-04296-t002:** ANOVA of the mixture design for the response variable: viability (log_10_ CFU/mL).

	Sum of Squares	Degrees of Freedom	*p*-Value
Model	0.751	5	0.003
Linear	0.126	2	0.078
AB	0.491	1	0.005
AC	0.359	1	0.001
BC	0.001	1	0.834
Residuals	0.190	10	
Lack of fit	0.149	6	0.208
Pure Error	0.041	4	
R^2^	0.80		

## Data Availability

The data presented in this study are available on request from the corresponding author.

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
