# Peer review of "Probiotic Encapsulation: Bead Design Improves Bacterial Performance during In Vitro Digestion"

_polymers, 2023, doi:10.3390/polym15214296_

Round 1

Reviewer 1 Report

The manuscript " Probiotic Encapsulation: bead Design Improves Bacterial Per-2 formance During In Vitro Digestion" demonstrated a novel method to encapsulate the Lactobacillus fermentum K73 by ionotropic gelation. Overall, the studies appear to be done well and the data are clearly presented. However, there are minor points that the authors should address before further consideration for publication:

1. Introduction: Further research on the biological function of Lactobacillus fermentum K73 is needed.

2. Keywords: functional products is not a keyword.

3. Materials and Methods: Further experiment is needed to measure the particle size using professional instruments.

Author Response

Ms. Kiara Qi

Assistant Editor

MDPI Polymers Editorial Office

Dear Editor:

We are pleased to submit the revised version of the polymers-2649740 manuscript that is: “Probiotic encapsulation: bead design improves bacterial performance during in vitro digestion. We appreciate the constructive criticisms of the reviewers. We have addressed each of their observations as described below. All the answers are in italics.

Reviewer 1

 Introduction: Further research on the biological function of Lactobacillus fermentum K73 is needed

The manuscript highlights several references related to this topic. Lactobacillus fermentum K73, a hypocholesterolemic probiotic strain, was isolated from a food product on the Colombian coast at the University of La Sabana. This strain has piqued the interest of the Agroindustrial Process Research Group (GIPA).

The studies conducted have centered on evaluating its probiotic potential, as reported by Cueto & Aragón (2012). Following the isolation process, various assessments were made, including its tolerance to gastric juice and bile salts, resistance to antibiotics of epidemiological importance, quantification of the activity of the enzyme bile salt hydrolase (BSH), and the adsorption of cholesterol. Additionally, the fermentation process has been explored by designing a culture medium and examining operational variables such as pH and agitation speed (Aragon et al., 2018). Similar investigations were conducted in different encapsulation processes (Aragon et al., 2019, Aragon et al., 2020).

  • Keywords: ‘functional products’ is not a keyword.

Thank you for your feedback. We have removed "functional products" as a keyword and, instead, we are suggesting "functional food" as replacement.

  • Materials and Methods: Further experiment is needed to measure the particle size using professional instruments.

Particle size of Ca (II)-alginate beads are usually measure by digital image analysis (Zazzali et al., 2019, Santagapita et al., 2012, Qi et al., 2020, Comunian & Favaro-Trindad, 2016). Given the relatively "large size" of the beads of this study it cannot be effectively measured using a zetasizer based on dynamic light scattering. Additional references related to this technique are provided in the manuscript.

 References:

  1. Cueto, C., & Aragón, S. (2012). Evaluation of probiotic potential of lactic acid bacteria to reduce in vitro cholesterol. Scientia Agropecuaria, 1(1), 45–50. https://doi.org/hpptts: //doi.org/10.17268/sci.agropecu.2012.01.06
  2. Stephanía Aragón-Rojas, Ruth Y. Ruiz-Pardo, Humberto Hernández-Sánchez & María Ximena Quintanilla-Carvajal (2018) Optimization of the production and stress resistance of the probiotic Lactobacillus fermentum K73 in a submerged bioreactor using a whey-based culture medium, CyTA - Journal of Food, 16:1, 1064-1070, DOI: 10.1080/19476337.2018.1527785
  3. Aragón Rojas, S., Quintanilla Carvajal, M. X., & Hernández Sánchez, H. (2018). Multifunctional Role of the Whey Culture Medium in the Spray Drying Microencapsulation of Lactic Acid Bacteria. 5417, 0–3. https://doi.org/10.17113/ftb.56.03.18.5285
  4. Stephania Aragón-Rojas, Ruth Yolanda Ruiz-Pardo, Alan Javier Hernández-Álvarez & María Ximena Quintanilla-Carvajal (2020) Sublimation conditions as critical factors during freeze-dried probiotic powder production, Drying Technology, 38:3, 333-349, DOI: 10.1080/07373937.2019.1570248
  5. Aragón-Rojas, S., Quintanilla-Carvajal, M.X., Hernández-Sánchez, H. et al. Encapsulation of Lactobacillus fermentum K73 by Refractance Window drying. Sci Rep 9, 5625 (2019). https://doi.org/10.1038/s41598-019-42016-0
  6. Santagapita, P. R., Mazzobre, M. F., & Buera, P. (2012). Invertase stability in alginate beads Effect of trehalose and chitosan inclusion and of drying methods. FRIN, 47(2), 321–330. https://doi.org/10.1016/j.foodres.2011.07.042
  7. Zazzali, I., Rocio, T., Calvo, A., Manuel, V., Ruíz-henestrosa, P., Santagapita, P. R., & Perullini, M. (2019). E ff ects of pH , extrusion tip size and storage protocol on the structural properties of Ca ( II ) -alginate beads. Carbohydrate Polymers, 206(November 2018), 749–756. https://doi.org/10.1016/j.carbpol.2018.11.051
  8. Qi, X., Simsek, S., Chen, B., & Rao, J. (2020). International Journal of Biological Macromolecules Alginate-based double-network hydrogel improves the viability of encapsulated probiotics during simulated sequential gastrointestinal digestion : Effect of biopolymer type and concentrations. International Journal of Biological Macromolecules, 165, 1675–1685. https://doi.org/10.1016/j.ijbiomac.2020.10.028
  9. Comunian, Talita & Favaro-Trindade, Carmen. (2016). Microencapsulation using biopolymers as an alternative to produce food enhanced with phytosterols and omega-3 fatty acids: A review. Food Hydrocolloids. 61. 10.1016/j.foodhyd.2016.06.003.

We thank you once more for your valuable observations.

Sincerely,

The authors

Reviewer 2 Report

Title at 2.5.2 should be rewritten "Scanning Electron Microscopy"

Figure 3A: the scale of the figure should be inserted

The authors used air and freeze drying for encapsulated product. The SEM in figure 3 was shown the deep crack on the beads. Have these cracks had any effect on the survival of bacteria?

Author Response

 Ms. Kiara Qi

Assistant Editor

MDPI Polymers Editorial Office

Dear Editor:

We are pleased to submit the revised version of the polymers-2649740 manuscript that is: “Probiotic encapsulation: bead design improves bacterial performance during in vitro digestion. We appreciate the constructive criticisms of the reviewers. We have addressed each of their observations as described below. All the answers are in italics.

Reviewer 2

  • Title at 2.5.2 should be rewritten "Scanning Electron Microscopy"

Thank you for the comment, the correction was done. Line 182.

  • Figure 3A: the scale of the figure should be inserted

We appreciate the suggestion; the scale bar was added to figure 3A.

  • The authors used air and freeze drying for encapsulated product. The SEM in figure 3 was shown the deep crack on the beads. Have these cracks had any effect on the survival of bacteria?

Thanks for your question. In our study, we exposed the beads to both freeze-drying and air drying to assess the morphological changes resulting from these drying methods. It's important to note that we did not include viability as a measured variable for the dried beads; all our assays were conducted using fresh beads. To address these questions regarding the impact of drying methods on cell viability and determine the most suitable drying approach for extending the shelf life of the beads without significant losses in cell viability, further studies will be necessary.

We thank you once more for your valuable observations.

Sincerely,

The authors